# Lifetime Spousal Violence Victimization and Perpetration, Physical Illness, and Health Risk Behaviours among Women in India

**DOI:** 10.3390/ijerph15122737

**Published:** 2018-12-04

**Authors:** Supa Pengpid, Karl Peltzer

**Affiliations:** 1ASEAN Institute for Health Development, Mahidol University, Salaya 73170, Thailand; supaprom@yahoo.com; 2Department of Research & Innovation, University of Limpopo, Turfloop 0727, South Africa; 3HIV/AIDS/STIs and TB (HAST), Human Sciences Research Council, Pretoria 0002, South Africa

**Keywords:** lifetime spousal violence, victimization, perpetration, women, health outcomes, India

## Abstract

The aim of this study was to assess the association between lifetime spousal violence victimization, spousal violence perpetration, and physical health outcomes and behaviours among women in India. In the 2015–2016 National Family Health Survey, a sample of ever-married women (15–49 years) (*N* = 66,013) were interviewed about spousal violence. Results indicate that 29.9% of women reported lifetime spousal physical violence victimization and 7.1% lifetime spousal sexual violence victimization (31.1% physical and/or sexual violence victimization), and 3.5% lifetime spousal physical violence perpetration. Lifetime spousal violence victimization and lifetime spousal violence perpetration were significantly positively correlated with asthma, genital discharge, genital sores or ulcers, sexually transmitted infections (STIs), tobacco use, alcohol use, and termination of pregnancy, and negatively associated with daily consumption of dark vegetables. In addition, lifetime spousal violence victimization was positively associated with being underweight, high random blood glucose levels, and anaemia, and negatively correlated with being overweight or obese. Lifetime spousal violence perpetration was marginally significantly associated with hypertension. The study found in a national sample of women in India a decrease of lifetime physical and/or sexual spousal violence victimization and an increase of lifetime spousal physical violence perpetration from 2005/5 to 2015/6. The results support other studies that found that, among women, lifetime spousal physical and/or sexual spousal violence victimization and lifetime spousal physical violence perpetration increase the odds of chronic conditions, physical illnesses, and health risk behaviours.

## 1. Introduction

Spousal or intimate partner violence can be defined as behaviour by a spouse, ex-spouse, ex-partner, or current partner that causes physical, sexual or psychological harm [1]. A spouse or intimate partner can experience spousal or intimate partner violence as a victim and/or can perpetrate spousal violence [2]. Spousal or intimate partner violence has been identified as an important public health problem, including in Asian countries [2,3]. In the 2005–2006 National Family Health Survey in India, among ever-married women (15–49 years old), the lifetime prevalence of less severe and severe physical spousal violence victimization was 31% and 10%, respectively, and lifetime spousal sexual violence victimization was 8% [4]. In a systematic review of spousal violence victimization in India, the lifetime prevalence was 41% [5]. There is a lack of more recent data on spousal lifetime violence victimization and no data on lifetime spousal violence perpetration and its correlates with health outcomes in India.

Having ever experienced intimate partner violence has been associated with a higher prevalence of chronic diseases [6], such as asthma [7,8], type 2 diabetes [9], cancer [8,10], anaemia [11], and other physical illnesses, such as sexually transmitted infections (STIs) [12,13,14] and reproductive tract infections (genital sores, abnormal genital discharge, etc.) [8,13,15]. According to a systematic review, intimate partner violence perpetration may be associated with a higher risk of cardiovascular risk and disease (including greater systolic blood pressure, incident hypertension, and self-report cardiac disease) [16]. Clark et al. [17] found in a longitudinal study that men with severe partner violence perpetration and victimization developed a higher rate of incident hypertension.

Other health risks, such as low body mass index (BMI) [11,18] and obesity [19,20,21,22], were found at a higher rate in women who had experienced lifetime intimate partner violence compared to women who had never experienced intimate partner violence. A number of studies found that intimate partner violence victimization was associated with various health compromising behaviours, such as tobacco use [7,13,21,22,23,24], (heavy) alcohol use [7,22,23,24], eating unhealthy foods [25], and termination of pregnancy [26]. The aim of this study was to assess the association between lifetime spousal violence victimization, spousal violence perpetration, and physical health outcomes and behaviours among women in India.

## 2. Materials and Methods

### 2.1. Sample and Procedure

Women aged 15 to 49 years (*N* = 66,013, individual response rate 94.5%) participated in the 2015–2016 India National Family Health Survey (NFHS-4) [27]. The NFHS-4 employed a two stage stratified sampling design [27]. The data in this study were restricted to a sub-sample of ever-married women that responded to the domestic violence questions (*N* = 66,013) of the NFHS-4. Prior to the study, informed consent was obtained from the study participants. The respective ethics committees of the participating institutions that implemented the NFHS-4 approved the study protocol. Permission to use the NFHS-4 data in this analysis was obtained from the Demographic and Health Surveys (DHS) Programme.

### 2.2. Measures

*Sociodemographic variables* included age, formal education, wealth status, number of living children, residence, religion, and caste [27]. Specific ethnic groups are categorized into scheduled castes, scheduled tribes, and other backward classes [4]. These groups are entitled to positive discrimination in terms of developmental opportunities [4]. Besides these castes (uncategorized), most of the population can be categorized as forward castes [4].

Physical violence by the (last) husband included the exposure to one or more of seven items, e.g., “Push you, shake you, or throw something at you?” and “Did any previous husband ever hit, slap, kick, or do anything else to hurt you physically?” [27].

Sexual spousal violence victimization by the (last) husband or former husband included exposure to one or more of three items, e.g., “Physically force you to have sexual intercourse with him even when you did not want to?” and “Did any previous husband physically force you to have intercourse or perform any other sexual acts against your will?” [27].

Physical spousal violence perpetration: “Have you ever hit, slapped, kicked, or done anything else to physically hurt your (last) husband at times when he was not already beating or physically hurting you?” [27].

*Anthropometry:* “Height and weight of adult women were measured using the Seca 874 digital scale” [27]. “Body mass index (BMI) was calculated according to Asian criteria: underweight (<18.5 kg/m^2^), normal weight (18.5 to <23.0 kg/m^2^), overweight (23.0 to <25.0 kg/m^2^) and obese (≥25 kg/m^2^)” [28].

*Blood pressure measurement:* “Blood pressure was measured using an Omron Blood Pressure Monitor. Blood pressure measurements for each respondent were taken three times with an interval of five minutes between readings. Respondents whose average systolic blood pressure (SBP) was >140 mm Hg or average diastolic blood pressure (DBP) was >90 mm Hg and/or were taking anti-hypertensive medication were considered to have hypertension” [27].

*Blood glucose testing:* “Random blood glucose (RBS) was measured using a finger-stick blood specimen for using the FreeStyle Optium H glucometer with glucose test strips” [27].

*Anaemia testing:* “Blood samples for anaemia testing were drawn from a drop of blood taken from a finger prick and collected in a microcuvette. Haemoglobin analysis was conducted on-site with a battery-operated portable HemoCue Hb 201+ analyser” [27]. “Anaemia was defined as haemoglobin level in <11.0 g/decilitre in non-pregnant and <12.0 in pregnant women aged 15–49 years. Haemoglobin levels are adjusted for smoking, and for altitude in enumeration areas that are above 1000 metres” [27].

*Other health issues* assessed by structured interview included tobacco and alcohol use, current morbidity (asthma, heart disease, cancer), termination of pregnancy, fruit and vegetable consumption, sexually transmitted infection, genital sore or ulcer, and bad smelling abnormal genital discharge in the past 12 months [27].

### 2.3. Data Analysis

Data were analysed with STATA software version 15.0 (Stata Corporation, College Station, Texas, USA) by considering the multi-stage study design. Descriptive statistics were used to describe the prevalence of spousal violence and the sample characteristics. Chi-square tests were used to calculate differences in proportions. Multinomial logistic regression was conducted to assess associations between independent variables (sociodemographic factors and violence related variables) and the dependent variables of being underweight or overweight/obese (with normal body weight status as reference category) and high and very high random glucose (with normal random blood glucose as reference category). Multivariable logistic regression analyses were used to calculate the association between independent variables (sociodemographic factors and violence related variables) and dependent variables, including each health outcome (health risks or diseases and health risk behaviours). *P* < 0.05 was regarded as statistically significant. 

## 3. Results

### 3.1. Sociodemographic Sample Characteristics

Overall, 66,013 women (median age 33 years, Interquartile Range= IQR: 14 years) responded to the domestic violence module of the 2015–2016 NFHS-4. More than half of the women (53.2%) had secondary or higher education, 89.6% had one or more living children, and 65.3% were living in rural areas. Most women (81.5%) were Hindu by religion, and 46.2% belonged to other backward classes (see Table 1).

### 3.2. Health and Violence Sample Characteristics

In all, 29.9% of women reported lifetime spousal physical violence victimization and 7.1% lifetime spousal sexual violence victimization (31.1% physical and/or sexual violence victimization), and 3.5% lifetime spousal physical violence perpetration. The prevalence of anaemia was 53.8%, being underweight 18.1%, being overweight or obese 39.5%, elevated random blood glucose 6.8%, hypertension 13.7%, heart disease 1.6%, asthma 2.0%, cancer 0.2%, and past 12-month sexually transmitted infection 2.4%. Regarding health risk behaviour, 7.1% of the women were using tobacco, 1.4% drank alcohol, 12.4% had fruits daily, 48.2% had dark vegetables daily, and 16.3% had ever terminated a pregnancy. In bivariate analyses, lifetime spousal violence victimization was associated with all health risks, physical illnesses, and health risk behaviours, except for hypertension, while lifetime spousal violence perpetration was associated with six out of ten health risks or diseases and all five health risk behaviours (see Table 2).

### 3.3. Associations with Health Outcomes

Table 3 depicts the results of adjusted logistic regression models predicting various health risks, diseases, and health risk behaviours for women with lifetime spousal violence victimization and lifetime spousal violence perpetration, separately. Lifetime spousal violence victimization and lifetime spousal violence perpetration were significantly positively correlated with asthma, genital discharge, genital sores or ulcers, STI, tobacco use, alcohol use, and termination of pregnancy, and negatively associated with daily consumption of dark vegetables. In addition, lifetime spousal violence victimization was positively associated with being underweight, high random blood glucose levels, and anaemia, and negatively correlated with being overweight or obese. Lifetime spousal violence perpetration was marginally significantly associated with hypertension (see Table 3).

## 4. Discussion

In the 2015–2016 India National Family Health Survey (NFHS-4), a prevalence of 31.1% of lifetime spousal physical and/or sexual violence victimization was found, which seemed to indicate a decline compared to the 2005–2006 India National Family Health Survey (NFHS-3) (37%) [27]. The study found a lifetime spousal partner violence perpetration of 3.5%, which seems to be an increase compared to the 2005–2006 NFHS-3 (1.8%) [29]. The reduction in spousal violence victimization may be due to the introduction of numerous policies, laws, and programmes, such as government-run helpline, crisis centers, and shelters for women who have experienced violence, by the Indian government to eliminate violence against women and girls [30]. The increase in spousal partner violence perpetration may be attributed to a greater divergence of traditional gender norms, for example in household decision-making power [31].

In agreement with previous studies [6,7,8,9,10,11], this study found that lifetime spousal violence victimization was associated with a higher prevalence of chronic conditions, including asthma, cancer, high glucose levels, and anaemia. Possible mechanisms for the relationship between domestic violence and anemia include “withholding of food as a form of abuse and stress-mediated influences of domestic violence on nutritional outcomes” [11]. The mechanism by which domestic violence affects cancer may be indirect through psychosocial stress or negative coping behaviours [32].

Consistent with previous studies in India [11,18], this study found an association between lifetime spousal violence victimization and being underweight. Possible explanations for the relationship between spousal violence victimization and nutritional deficiencies may include the withholding of food and a mediating effect of psychological distress that could trigger weight loss [33,34]. While previous studies in western countries and in Saudi Arabia [19,20,21,22] found that having experienced intimate partner violence among women increased the odds for obesity, this study found that lifetime spousal physical and/or sexual violence victimization among women decreased the odds of being overweight or obese. This possible cultural difference needs further investigations. Previous studies [16,17] found a relationship between intimate partner violence perpetration and a higher risk of cardiovascular risk and disease (including incident hypertension and self-reported cardiac disease), while this study found associations between lifetime spousal violence victimization and heart disease and, in bivariate analysis, between lifetime spousal physical violence perpetration and hypertension. “Intimate partner violence could be a risk marker for maladaptive stress responses that lead to cardiovascular events” [16]. No association was found between lifetime spousal violence victimization and measured hypertension, which is consistent with a previous review [35] on intimate partner violence and measured hypertension.

Furthermore, confirming results from a number of studies [8,12,13,14,15], this study found that lifetime spousal violence victimization and lifetime spousal violence perpetration were significantly positively correlated with genital discharge, genital sores or ulcers, and STIs. Consistent with several previous studies [7,13,21,22,23,24], this study found that lifetime spousal violence victimization and lifetime spousal violence perpetration were associated with tobacco and alcohol use. Some investigators propose that, in many cases, intimate partner violence precedes alcohol and substance use, these substances subsequently being used as coping mechanisms against violent experiences [23,24]. As this is a cross-sectional study, the direction of the intimate partner violence and substance use relationship cannot be established; this should be done in longitudinal studies.

Moreover, this study found an association between both lifetime spousal violence victimization and lifetime spousal violence perpetration and eating unhealthy foods (less than daily fruit and vegetable consumption), which has also been found in one previous study [25]. Some researchers have suggested that in addition to substance use, such as tobacco use, unhealthy dietary behaviours could be adverse coping strategies to cope with intimate partner violence induced stress [7,21]. Consistent with a previous study [26], this study found that both lifetime spousal violence victimization and lifetime spousal violence perpetration were associated with termination of pregnancy. Women in this study may benefit from health care interventions that integrate stopping violence and promoting a healthy lifestyle [21]. Furthermore, health care providers can screen women for domestic violence and provide early and appropriate support and physical and mental health care [36].

### Study Limitations

This study was cross-sectional, so no causative conclusions can be drawn. Future investigations should also assess spousal violence victimization and perpetration, including mental health outcomes and including men in India.

## 5. Conclusions

The study found in a national sample of women in India a decrease in lifetime physical and/or sexual spousal violence victimization and an increase in lifetime spousal physical violence perpetration from 2005/5 to 2015/6. Our results support other studies that found among women lifetime spousal physical and/or sexual spousal violence victimization, and lifetime spousal physical violence perpetration, increase the odds of chronic conditions, physical illnesses, and health risk behaviours. Women who have experienced and/or perpetrated spousal violence may be targeted for the prevention of chronic and physical illnesses, such as being underweight, anaemia, heart disease, asthma, cancer, and sexually transmitted and reproductive tract infections, as well as health risk behaviours, such as substance use, unhealthy diet, and termination of pregnancy.

## Figures and Tables

**Table 1 ijerph-15-02737-t001:** Sociodemographic sample characteristics of the domestic violence module of the India National Family Health Survey (NFHS-4) (*N* = 66,013).

Variable	*N*	%
Age		
15–29	24,462	37.8
30–39	24,998	34.3
40–49	16,553	27.9
Education		
None	22,028	32.5
Primary	9669	14.3
Secondary	28,187	42.8
Higher	6129	10.4
Number of living children		
0	6136	10.4
1–3	49,189	74.0
4 or more	10,688	15.6
Wealth status		
Poorest	12,838	17.0
Poorer	13,992	19.3
Middle	13,790	20.7
Richer	13,142	21.2
Richest	12,251	21.7
Rural residence	46,544	65.3
Urban residence	19,469	34.7
Religion		
Hindu	49,546	81.5
Muslim	8614	13.7
Other or none	7123	4.8
Caste		
Scheduled caste	11,686	20.3
Scheduled tribe	12,108	9.6
Other backward class	25,574	46.2
Other	13,449	23.9

**Table 2 ijerph-15-02737-t002:** Sample description by health and violence variables.

Variable	Variable Response Options	Sample (*N* = 66,013)	Physical and/or Sexual Violence Victimization (*n* = 19,561)		Physical Violence Perpetration (*n* = 2128)	
		*N* (%)	%	Chi-square	%	Chi-square
*All*			31.1	*p*-value	3.5	*p*-value
*Health Risk/Disease*						
	No	30,552 (46.2)	52.5		53.7	
Anaemia	Yes	33,938 (53.8)	56.6	<0.001	57.3	0.030
	Missing	1523				
Body mass index	Normal weight	29,032 (42.4)	43.9		42.9	
	Underweight	11,607 (18.1)	21.1	<0.001	19.7	0.286
	Overweight or obese	24,212 (39.5)	35.1		37.5	
	Missing	1162				
	No	53,760 (86.3)	14.0		13.6	
Hypertension	Yes	8770 (13.7)	13.1	0.193	16.3	0.015
	Missing	3483				
Random blood glucose	Normal	60,267 (93.2)	92.9		92.7	
	High (141–160 mg/dL)	2213 (3.4)	3.8	0.040	3.4	0.757
	V/high (>160 mg/dL)	2008 (3.4)	3.3		3.9	
	Missing	1525				
Heart disease	No Yes	64,530 (98.4) 1097 (1.6)	1.2 2.3	<0.001	1.5 2.0	0.202
Asthma	No Yes	64,572 (98.0) 1107 (2.0)	1.5 3.0	<0.001	1.9 4.0	<0.001
Cancer	No Yes	65,571 (99.8) 77 (0.2)	0.1 0.4	0.016	0.2 0.3	0.431
Genital discharge (past 12 months)	No Yes	58,288 (90.7) 6979 (9.3)	7.4 13.7	<0.001	9.0 17.5	<0.001
Genital sore/ulcer (past 12 months)	No Yes	63,229 (97.1) 2058 (2.9)	2.3 4.4	<0.001	2.7 9.1	<0.001
STI (past 12 months)	No Yes	64,331 (97.6) 1607 (2.4)	2.3 2.8	0.008	2.4 4.3	<0.001
Terminated pregnancy	No Yes	55,089 (83.7) 10,924 (16.3)	14.5 20.3	<0.001	16.1 20.8	<0.001
*Health risk behaviour*						
Tobacco use	No Yes	59,216 (92.9) 6797 (7.1)	5.6 10.5	<0.001	6.9 12.2	<0.001
Drinks alcohol	No Yes	64,077 (98.6) 1936 (1.4)	0.9 2.6	<0.001	1.4 3.1	<0.001
Fruits daily	No Yes	58,621 (87.6) 7392 (12.4)	14.8 6.9	<0.001	12.5 10.3	0.055
Dark vegetables daily	No Yes	33,445 (51.8) 32,568 (48.2)	50.0 44.0	<0.001	48.4 41.4	<0.001

**Table 3 ijerph-15-02737-t003:** Multivariable risk or odds ratios (with 95% confidence intervals) for health outcomes. Predictors: Physical and/or sexual violence victimization and physical violence perpetration (model *N* = 66,212).

Variable	Physical and/or Sexual Violence Victimization		Physical Violence Perpetration	
*Health risk/disease*				
	ARRR (95% CI) ^1^	*P*-value	ARRR (95% CI) ^1^	*P*-value
Body mass index			---	
Normal weight	1 (Reference)			
Underweight	1.11 (1.04, 1.19)	0.002		
Overweight or obese	0.92 (0.86, 0.98)	0.011		
Random blood glucose			---	
Normal	1 (Reference)			
High (141–160 mg/dl)	1.16 (1.00, 1.35)	0.047		
Very high (>160 mg/dl)	0.90 (0.77, 1.06)	0.219		
	AOR (95% CI) ^1^		AOR (95% CI) ^1^	
Anaemia	1.09 (1.03, 1.15)	0.002	1.10 (0.96, 1.26)	0.167
Hypertension	---		1.19 (0.98, 1.42)	0.080
Heart disease	1.88 (1.54, 2.30)	<0.001	---	
Asthma	2.04 (1.60, 2.49)	<0.001	2.15 (1.51, 3.07)	<0.001
Cancer	3.62 (1.55, 8.46)	0.003	---	
Genital discharge (past 12 months)	1.92 (1.76, 2.10)	<0.001	2.13 (1.78, 2.54)	<0.001
Genital sore/ulcer (past 12 months)	2.07 (1.75, 2.45)	<0.001	3.67 (2.80, 4.81)	<0.001
STI (past 12 months)	1.37 (1.14, 1.65)	<0.001	1.94 (1.43, 2.63)	<0.001
Terminated pregnancy	1.54 (1.44, 1.66)	<0.001	1.39 (1.18, 1.63)	<0.001
*Health risk behaviour*				
Tobacco use	1.43 (1.29, 1.57)	<0.001	1.60 (1.33, 1.93)	<0.001
Drinks alcohol	2.16 (1.78, 2.64)	<0.001	1.88 (1.33, 2.66)	<0.001
Fruits daily (base = less than daily)	0.63 (0.56, 0.71)	<0.001	---	
Dark vegetables daily (base = less than daily)	0.83 (0.78, 0.89)	<0.001	0.78 (0.68, 0.90)	<0.001

ARRR = Adjusted Relative Risk Ratio; AOR = Adjusted Odds Ratio; ^1^ Adjusted for age, education, wealth status, number of living children, rural-urban, religion and caste; STI = Sexually transmitted infection.

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
