# Peer review of "Lifetime Spousal Violence Victimization and Perpetration, Physical Illness, and Health Risk Behaviours among Women in India"

_ijerph, 2018, doi:10.3390/ijerph15122737_

Round 1
Reviewer 1 Report
Line 49 - needs the word "and"; Line 84 - typo maybe - should it be or rather than of?
Line 116 - in the sample characteristics, it may be useful to define what is meant by the different caste levels, just for those of us who are unfamiliar with what the different categories mean. Lines 152-156 are redundant - you've made that information clear in a number of other places so feel free to eliminate it here. Lines 181-182 - should just read "cannot"...currently reads "cannot not". Lines 188-189, it is unclear what is being stated. In the discussion/conclusion, could there be more discussion on referral to a public health program or victim advocacy program dealing with recovery from domestic violence, dealing with current domestic violence, and healthy lifestyles? Attaching an additional article you may want to integrate to assist with conclusions.
Overall - an incredibly valuable piece. The impact of intimate partner violence, particularly the effect of stress on the immune system and general physical health, is something many are uncomfortable discussing. However, it offers us the ability to address recovery in a more holistic manner, not just focusing on psychological recovery, or recovery of a broken bone, but also of the long-term physical health deterioration that may be experienced by victims of domestic violence. Truly valuable research.

Author Response
Reviewer 1:
Line 49 - needs the word "and"; Line 84 - typo maybe - should it be or rather than of?
Response: Corrected
Line 116 - in the sample characteristics, it may be useful to define what is meant by the different caste levels, just for those of us who are unfamiliar with what the different categories mean.
Response: below is added
Specific ethnic groups are categorized into scheduled castes, scheduled tribes, and other backward classes [4]. These groups are entitled to positive discrimination in terms of developmental opportunities [4]. Besides these castes (uncategorized), most of the population can be categorized as forward castes [4].
Lines 152-156 are redundant - you've made that information clear in a number of other places so feel free to eliminate it here.
Response: Needs to be discussed
Lines 181-182 - should just read "cannot"...currently reads "cannot not".
Response: Corrected
Lines 188-189, it is unclear what is being stated.
Response: corrected as below
Consistent with a previous study [26], this study found that both lifetime spousal violence victimization and lifetime spousal violence perpetration were associated with termination of pregnancy
In the discussion/conclusion, could there be more discussion on referral to a public health program or victim advocacy program dealing with recovery from domestic violence, dealing with current domestic violence, and healthy lifestyles? Attaching an additional article you may want to integrate to assist with conclusions.
Response: below is added
Women in this study may benefit from health care interventions that integrates stopping violence and promoting a healthy lifestyle [21]. Further, health care providers can screen women for domestic violence and provide early and appropriate support and physical and mental health care [31].
1. Asadi, S.; Mirghafourvand, M.; Yavarikia, P.; Mohammad-Alizadeh-Charandabi, S.; Nikan, F. Domestic violence and its relationship with quality of life in Iranian women of reproductive age. J Fam Viol. 2017; 32:453-460. DOI: 10.1007/s10896-016-9832-0
Reviewer 2 Report
This paper provides valuable data on the relationship between intimate partner violence and chronic health issues. This goes beyond the well established physical harm that is caused by injury and can inform health professionals to consider harm beyond the initial impact.
The paper presents extensive data in quite a dense way and some further discussion of some of the key relationships would be useful. The comments below are organized by line and include significant and also minor points.
35 lists spouse, ex-spouse and ex-partner but omits current partner.
43 "There are lack of" should be "there is a lack of"
53 "men experiencing severe perpetration" suggests that being violent is something that happens to men. Reword.
59-60 (and Table 2) The categorization of termination of pregnancy (which I take to mean abortion) as a health compromising or health risk behavior is troubling. It may in fact be a choice made to protect the woman's health. Pregnancy and child birth pose greater risks than abortion and pregnancy is associated with partner violence which women are trying to avoid.
75 The section describing the violence variables should have a separate heading not be listed under socio-demographic vars.
122 Table 1 It would be helpful to explain the caste/tribe categories for an international readership that may not be familiar with these terms. I suppose that the term "Other backward class" (which is the modal category) is acceptable usage in India. To an international reader it is a demeaning label that undermines the dignity of those so identified. Please find a more neutral term.
152 The Discussion session describes a reduction in victimization rate and an increase in perpetration rate over the decade. Some discussion of why this shift has a occurred would be useful.
In general, this paper stays very close to its data and it is correct that correlational data does not establish causality. However, I would urge the authors to provide more potential explanation for the valuable findings they present. They know these data best and more of their analysis would be welcome.
Author Response
Reviewer 2
This paper provides valuable data on the relationship between intimate partner violence and chronic health issues. This goes beyond the well established physical harm that is caused by injury and can inform health professionals to consider harm beyond the initial impact.
The paper presents extensive data in quite a dense way and some further discussion of some of the key relationships would be useful. The comments below are organized by line and include significant and also minor points.
35 lists spouse, ex-spouse and ex-partner but omits current partner.
Response: added
43 "There are lack of" should be "there is a lack of"
Response: “data” is plural
53 "men experiencing severe perpetration" suggests that being violent is something that happens to men. Reword.
Response: Modified
59-60 (and Table 2) The categorization of termination of pregnancy (which I take to mean abortion) as a health compromising or health risk behavior is troubling. It may in fact be a choice made to protect the woman's health. Pregnancy and child birth pose greater risks than abortion and pregnancy is associated with partner violence which women are trying to avoid.
Response: Modified
75 The section describing the violence variables should have a separate heading not be listed under socio-demographic vars.
Response: Modified
122 Table 1 It would be helpful to explain the caste/tribe categories for an international readership that may not be familiar with these terms. I suppose that the term "Other backward class" (which is the modal category) is acceptable usage in India. To an international reader it is a demeaning label that undermines the dignity of those so identified. Please find a more neutral term.
Response: as below
Specific ethnic groups are categorized into scheduled castes, scheduled tribes, and other backward classes [4]. These groups are entitled to positive discrimination in terms of developmental opportunities [4]. Besides these castes (uncategorized), most of the population can be categorized as forward castes [4].
Used the term “other backward class” as used in international publications.
152 The Discussion session describes a reduction in victimization rate and an increase in perpetration rate over the decade. Some discussion of why this shift has a occurred would be useful.
Response: below is added
The reduction in spousal violence victimization may be due to the introduction of numerous policies, laws, and programmes, such as government-run helpline, crisis centers, and shelters for women who have experienced violence, by the Indian government to eliminate violence against women and girls [30]. The increase in spousal partner violence perpetration may be attributed to a greater divergence of traditional gender norms, for example in household decision-making power [31].
In general, this paper stays very close to its data and it is correct that correlational data does not establish causality. However, I would urge the authors to provide more potential explanation for the valuable findings they present. They know these data best and more of their analysis would be welcome.
Response: Several additions are added